# Powerset Convolutional Neural Networks

**Chris Wendler**
Department of Computer Science
ETH Zurich, Switzerland
chris.wendler@inf.ethz.ch

**Dan Alistarh**
IST Austria
dan.alistarh@ist.ac.at

**Markus Püschel**
Department of Computer Science
ETH Zurich, Switzerland
pueschel@inf.ethz.ch

## Abstract

We present a novel class of convolutional neural networks (CNNs) for set functions, i.e., data indexed with the powerset of a finite set. The convolutions are derived as linear, shift-equivariant functions for various notions of shifts on set functions. The framework is fundamentally different from graph convolutions based on the Laplacian, as it provides not one but several basic shifts, one for each element in the ground set. Prototypical experiments with several set function classification tasks on synthetic datasets and on datasets derived from real-world hypergraphs demonstrate the potential of our new powerset CNNs.

## 1   Introduction

Deep learning-based methods are providing state-of-the-art approaches for various image learning and natural language processing tasks, such as image classification [22, 28], object detection [41], semantic image segmentation [42], image synthesis [20], language translation / understanding [23, 62] and speech synthesis [58]. However, an artifact of many of these models is that regularity priors are hidden in their fundamental neural building blocks, which makes it impossible to apply them directly to irregular data domains. For instance, image convolutional neural networks (CNNs) are based on parametrized 2D convolutional filters with local support, while recurrent neural networks share model parameters across different time stamps. Both architectures share parameters in a way that exploits the symmetries of the underlying data domains.

In order to port deep learners to novel domains, the according parameter sharing schemes reflecting the symmetries in the target data have to be developed [40]. An example are neural architectures for graph data, i.e., data indexed by the vertices of a graph. Graph CNNs define graph convolutional layers by utilizing results from algebraic graph theory for the graph Laplacian [9, 51] and message passing neural networks [18, 47] generalize recurrent neural architectures from chain graphs to general graphs. With these building blocks in place, neural architectures for supervised [16, 18, 50], semi-supervised [25] and generative learning [52, 59] on graphs have been deployed. These research endeavors fall under the umbrella term of geometric deep learning (GDL) [10].

In this work, we want to open the door for deep learning on set functions, i.e., data indexed by the powerset of a finite set. There are (at least) three ways to do so. First, set functions can be viewed as data indexed by a hypercube graph, which makes graph neural nets applicable. Second, results from the Fourier analysis of set functions based on the Walsh-Hadamard-transform (WHT) [15, 33, 54] can be utilized to formulate a convolution for set functions in a similar way to [51]. Third, [36] introduces several novel notions of convolution for set functions (powerset convolution)

as linear, equivariant functions for different notions of shift on set functions. This derivation parallels the standard 2D-convolution (equivariant to translations) and graph convolutions (equivariant to the Laplacian or adjacency shift) [34]. A general theory for deriving new forms of convolutions, associated Fourier transforms and other signal processing tools is outlined in [38].

**Contributions** Motivated by the work on generalized convolutions and by the potential utility of deep learning on novel domains, we propose a method-driven approach for deep learning on irregular data domains and, in particular, set functions:

- We formulate novel powerset CNN architectures by integrating recent convolutions [36] and proposing novel pooling layers for set functions.

- As a protoypical application, we consider the set function classification task. Since there is little prior work in this area, we evaluate our powerset CNNs on three synthetic classification tasks (submodularity and spectral properties) and two classification tasks on data derived from real-world hypergraphs [5]. For the latter, we design classifiers to identify the origin of the extracted subhypergraph. To deal with hypergraph data, we introduce several set-function-based hypergraph representations.

- We validate our architectures experimentally, and show that they generally outperform the natural fully-connected and graph-convolutional baselines on a range of scenarios and hyperparameter values.

## 2   Convolutions on Set Functions

We introduce background and definitions for set functions and associated convolutions. For context and analogy, we first briefly review prior convolutions for 2D and graph data. From the signal processing perspective, 2D convolutions are linear, shift-invariant (or equivariant) functions on images $s : \mathbb{Z}^2 \to \mathbb{R}; (i,j) \mapsto s_{i,j}$, where the shifts are the translations $T_{(k,l)}s = (s_{i-k,j-l})_{i,j \in \mathbb{Z}^2}$. The 2D convolution thus becomes

$$(h * s)_{i,j} = \sum_{k,l \in \mathbb{Z}^2} h_{k,l} s_{i-k,j-l}. \tag{1}$$

Equivariance means that all convolutions commute with all shifts: $h * (T_{(k,l)}s) = T_{(k,l)}(h * s)$.

Convolutions on vertex-indexed graph signals $s : V \to \mathbb{R}; v \mapsto s_v$ are linear and equivariant with respect to the Laplacian shifts $T_k s = L^k s$, where $L$ is the graph Laplacian [51].

**Set functions** With this intuition in place, we now consider set functions. We fix a finite set $N = \{x_1, \dots, x_n\}$. An associated set function is a signal on the powerset of $N$:

$$s : 2^N \to \mathbb{R}; A \mapsto s_A. \tag{2}$$

**Powerset convolution** A convolution for set functions is obtained by specifying the shifts to which it is equivariant. The work in [36] specifies $T_Q s = (s_{A \setminus Q})_{A \subseteq N}$ as one possible choice of shifts for $Q \subseteq N$. Note that in this case the shift operators are parametrized by the monoid $(2^N, \cup)$, since for all $s$

$$T_Q(T_R s) = (s_{A \setminus R \setminus Q})_{A \subseteq N} = (s_{A \setminus (R \cup Q)})_{A \subseteq N} = T_{Q \cup R}s,$$

which implies $T_Q T_R = T_{Q \cup R}$. The corresponding linear, shift-equivariant *powerset convolution* is given by [36] as

$$(h * s)_A = \sum_{Q \subseteq N} h_Q s_{A \setminus Q}. \tag{3}$$

Note that the filter $h$ is itself a set function. Table 1 contains an overview of generalized convolutions and the associated shift operations to which they are equivariant to.

**Fourier transform** Each filter $h$ defines a linear operator $\Phi_h := (h * \cdot)$ obtained by fixing $h$ in (3). It is diagonalized by the powerset Fourier transform

$$F = \begin{pmatrix} 1 & 0 \\ 1 & -1 \end{pmatrix}^{\otimes n} = \begin{pmatrix} 1 & 0 \\ 1 & -1 \end{pmatrix} \otimes \cdots \otimes \begin{pmatrix} 1 & 0 \\ 1 & -1 \end{pmatrix}, \tag{4}$$

| | signal | shifted signal | convolution | reference | CNN |
|---|---|---|---|---|---|
| image | $(s_{i,j})_{i,j}$ | $(s_{i-k,j-l})_{i,j\in\mathbb{Z}}$ | $\sum_{k,l} h_{k,l} s_{i-k,j-l}$ | standard | standard |
| graph Laplacian | $(s_v)_{v\in V}$ | $(L^k s)_{v\in V}$ | $(\sum_k h_k L^k s)_v$ | [51] | [9] |
| graph adjacency | $(s_v)_{v\in V}$ | $(A^k s)_{v\in V}$ | $(\sum_k h_k A^k s)_v$ | [44] | [55] |
| group | $(s_g)_{g\in G}$ | $(s_{q^{-1}g})_{g\in G}$ | $\sum_q h_q s_{q^{-1}g}$ | [53] | [13] |
| group spherical | $(s_R)_{R\in\mathbf{SO}(3)}$ | $(s_{Q^{-1}R})_{R\in\mathbf{SO}(3)}$ | $\int h_Q s_{Q^{-1}R} d\mu(Q)$ | [12] | [12] |
| powerset | $(s_A)_{A\subseteq N}$ | $(s_{A\setminus Q})_{A\subseteq N}$ | $\sum_Q h_Q s_{A\setminus Q}$ | [36] | this paper |

Table 1: Generalized convolutions and their shifts.

where $\otimes$ denotes the Kronecker product. Note that $F^{-1} = F$ in this case and that the spectrum is also indexed by subsets $B \subseteq N$. In particular, we have

$$F\Phi_h F^{-1} = \text{diag}((\tilde{h}_B)_{B\subseteq N}), \tag{5}$$

in which $\tilde{h}$ denotes the frequency response of the filter $h$ [36]. We denote the linear mapping from $h$ to its frequency response $\tilde{h}$ by $\bar{F}$, i.e., $\tilde{h} = \bar{F}h$.

**Other shifts and convolutions** There are several other possible definitions of set shifts, each coming with its respective convolutions and Fourier transforms [36]. Two additional examples are $T_Q^\diamond s = (s_{A\cup Q})_{A\subseteq N}$ and the symmetric difference $T_Q^\bullet s = (s_{(A\setminus Q)\cup(Q\setminus A)})_{A\subseteq N}$ [54]. The associated convolutions are, respectively,

$$(h * s)_A = \sum_{Q\subseteq N} h_Q s_{A\cup Q} \quad \text{and} \quad (h * s)_A = \sum_{Q\subseteq N} h_Q s_{(A\setminus Q)\cup(Q\setminus A)}. \tag{6}$$

**Localized filters** Filters $h$ with $h_Q = 0$ for $|Q| > k$ are $k$-localized in the sense that the evaluation of $(h * s)_A$ only depends on evaluations of $s$ on sets differing by at most $k$ elements from $A$. In particular, 1-localized filters $(h * s)_A = h_\emptyset s_A + \sum_{x\in N} h_{\{x\}} s_{A\setminus\{x\}}$ are the counterpart of *one-hop* filters that are typically used in graph CNNs [25]. In contrast to the omnidirectional one-hop graph filters, these one-hop filters have one direction per element in $N$.

## 2.1 Applications of Set Functions

Set functions are of practical importance across a range of research fields. Several optimization tasks, such as cost effective sensor placement [27], optimal ad placement [19] and tasks such as semantic image segmentation [35], can be reduced to subset selection tasks, in which a set function determines the value of every subset and has to be maximized to find the best one. In combinatorial auctions, set functions can be used to describe bidding behavior. Each bidder is represented as a valuation function that maps each subset of goods to its subjective value to the customer [14]. Cooperative games are set functions [8]. A coalition is a subset of players and a coalition game assigns a value to every subset of players. In the simplest case the value one is assigned to winning and the value zero to losing coalitions. Further, graphs and hypergraphs also admit set function representations:

**Definition 1.** (Hypergraph) *A hypergraph is a triple* $H = (V, E, w)$*, where* $V = \{v_1, \ldots, v_n\}$ *is a set of vertices,* $E \subseteq (\mathcal{P}(V) \setminus \emptyset)$ *is a set of hyperedges and* $w : E \to \mathbb{R}$ *is a weight function.*

The weight function of a hypergraph is a set function on $V$ by setting $s_A = w_A$ if $A \in E$ and $s_A = 0$ otherwise. Additionally, hypergraphs induce two set functions, namely the hypergraph cut and association score function:

$$\text{cut}_A = \sum_{\substack{B\in E, B\cap A\neq\emptyset, \\ B\cap(V\setminus A)\neq\emptyset}} w_B \quad \text{and} \quad \text{assoc}_A = \sum_{B\in E, B\subseteq A} w_B. \tag{7}$$

## 2.2 Convolutional Pattern Matching

The powerset convolution in (3) raises the question of which patterns are "detected" by a filter $(h_Q)_{Q\subseteq N}$. In other words, to which signal does the filter $h$ respond strongest when evaluated at a

given subset $A$? We call this signal $p^A$ (the pattern matched at position $A$). Formally,

$$p^A = \arg\max_{s: \|s\|=1} (h * s)_A. \tag{8}$$

For $p^N$, the answer is $p^N = (1/\|h\|)(h_{N\setminus B})_{B \subseteq N}$. This is because the dot product $\langle h, s^* \rangle$, with $s_A^* = s_{N\setminus A}$, is maximal if $h$ and $s^*$ are aligned. Slightly rewriting (3) yields the answer for the general case $A \subseteq N$:

$$(h * s)_A = \sum_{Q \subseteq N} h_Q s_{A\setminus Q} = \sum_{Q_1 \subseteq A} \underbrace{\left( \sum_{Q_2 \subseteq N\setminus A} h_{Q_1 \cup Q_2} \right)}_{=:h'_{Q_1}} s_{A\setminus Q_1}. \tag{9}$$

Namely, (9) shows that the powerset convolution evaluated at position $A$ can be seen as the convolution of a new filter $h'$ with $s$ restricted to the powerset $2^A$ evaluated at position $A$, the case for which we know the answer: $p_B^A = (1/\|h'\|)h'_{A\setminus B}$ if $B \subseteq A$ and $p_B^A = 0$ otherwise.

**Example 1.** (One-hop patterns) *For a one-hop filter $h$, i.e., $(h * s)_A = h_\emptyset s_A + \sum_{x \in N} h_{\{x\}} s_{A\setminus\{x\}}$ the pattern matched at position $A$ takes the form*

$$p_B^A = \begin{cases} \frac{1}{\|h'\|}(h_\emptyset + \sum_{x \in N\setminus A} h_{\{x\}}) & \text{if } B = A, \\ \frac{1}{\|h'\|}h_{\{x\}} & \text{if } B = A \setminus \{x\} \text{ with } x \in A, \\ 0 & \text{else.} \end{cases} \tag{10}$$

*Here, $h'$ corresponds to the filter restricted to the powerset $2^A$ as in (9).*

Notice that this behavior is different from 1D and 2D convolutions: there the underlying shifts (translations) are invertible and thus the detected patterns are again shifted versions of each other. For example, the 1D convolutional filter $(h_q)_{q \in \mathbb{Z}}$ matches $p^0 = (h_{-q})_{q \in \mathbb{Z}}$ at position 0 and $p^t = T_{-t}p^0 = (h_{-q+t})_{q \in \mathbb{Z}}$ at position $t$, and, the group convolutional filter $(h_q)_{q \in G}$ matches $p^e = (h_{q^{-1}})_{q \in G}$ at the unit element $e$ and $p^g = T_{g^{-1}}p^e = (h_{gq^{-1}})_{q \in G}$ at position $g$. Since powerset shifts are not invertible, the detected patterns by a filter are not just (set-)shifted versions of each other as shown above.

A similar behavior can be expected with graph convolutions since the Laplacian shift is never invertible and the adjacency shift is not always invertible.

## 3  Powerset Convolutional Neural Networks

**Convolutional layers** We define a convolutional layer by extending the convolution to multiple channels, summing up the feature maps obtained by channel-wise convolution as in [10]:

**Definition 2.** (Powerset convolutional layer) *A powerset convolutional layer is defined as follows:*

1. *The input is given by $n_c$ set functions $\mathbf{s} = (s^{(1)}, \ldots, s^{(n_c)}) \in \mathbb{R}^{2^N \times n_c}$ ;*

2. *The output is given by $n_f$ set functions $\mathbf{t} = L_\Gamma(\mathbf{s}) = (t^{(1)}, \ldots, t^{(n_f)}) \in \mathbb{R}^{2^N \times n_f}$;*

3. *The layer applies a bank of set function filters $\Gamma = (h^{(i,j)})_{i,j}$, with $i \in \{1, \ldots, n_c\}$ and $j \in \{1, \ldots, n_f\}$, and a point-wise non-linearity $\sigma$ resulting in*

$$t_A^{(j)} = \sigma(\sum_{i=1}^{n_c} (h^{(i,j)} * s^{(i)})_A). \tag{11}$$

**Pooling layers** As in conventional CNNs, we define powerset pooling layers to gain additional robustness with respect to input perturbations, and to control the number of features extracted by the convolutional part of the powerset CNN. From a signal processing perspective, the crucial aspect of the pooling operation is that the pooled signal lives on a valid signal domain, i.e., a powerset. One way to achieve this is by combining elements of the ground set.

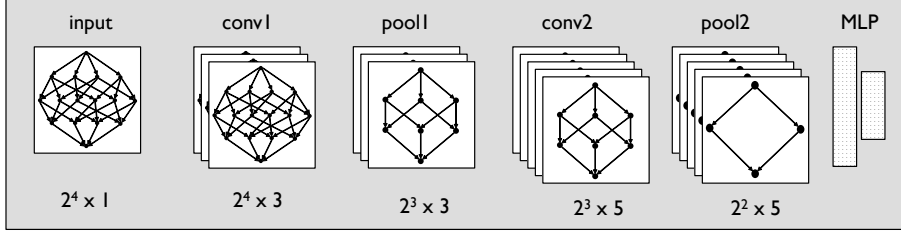

| input | conv1 | pool1 | conv2 | pool2 | MLP |
|---|---|---|---|---|---|
| $2^4 \times 1$ | $2^4 \times 3$ | $2^3 \times 3$ | $2^3 \times 5$ | $2^2 \times 5$ | |

Figure 1: Forward pass of a simple powerset CNN with two convolutional and two pooling layers. Set functions are depicted as signals on the powerset lattice.

**Definition 3.** (Powerset pooling) *Let $N'(X)$ be the ground set of size $n - |X| + 1$ obtained by combining all the elements in $X \subseteq N$ into a single element. E.g., for $X = \{x_1, x_2\}$ we get $N'(X) = \{\{x_1, x_2\}, x_3, \ldots, x_n\}$. Therefore every subset $X \subseteq N$ defines a pooling operation*

$$P^X : \mathbb{R}^{2^N} \to \mathbb{R}^{2^{N'(X)}} : (s_A)_{A \subseteq N} \mapsto (s_B)_{B : B \cap X = X \text{ or } B \cap X = \emptyset}. \tag{12}$$

In our experiments we always use $P := P^{\{x_1, x_2\}}$. It is also possible to pool a set function by combining elements of the powerset as in [48] or by the simple rule $s_B = \max(s_B, s_{B \cup \{x\}})$ for $B \subseteq N \setminus \{x\}$. Then, a pooling layer is obtained by applying our pooling strategy to every channel.

**Definition 4.** (Powerset pooling layer) *A powerset pooling layer takes $n_c$ set functions as input $\mathbf{s} = (s^{(1)}, \ldots, s^{(n_c)}) \in \mathbb{R}^{2^N \times n_c}$ and outputs $n_c$ pooled set functions $\mathbf{t} = L_P(\mathbf{s}) = (t^{(1)}, \ldots, t^{(n_c)}) \in \mathbb{R}^{2^{N'} \times n_c}$, with $|N'| = |N| - 1$, by applying the pooling operation to every channel*

$$t^{(i)} = P(s^{(i)}). \tag{13}$$

**Powerset CNN** A powerset CNN is a composition of several powerset convolutional and pooling layers. Depending on the task, the outputs of the convolutional component can be fed into a multi-layer perceptron, e.g., for classification.

Fig. 1 illustrates a forward pass of a powerset CNN with two convolutional layers, each of which is followed by a pooling layer. The first convolutional layer is parameterized by three one-hop filters and the second one is parameterized by fifteen (three times five) one-hop filters. The filter coefficients were initialized with random weights for this illustration.

**Implementation**[1] We implemented the powerset convolutional and pooling layers in Tensorflow [1]. Our implementation supports various definitions of powerset shifts, and utilizes the respective Fourier transforms to compute the convolutions in the frequency domain.

## 4 Experimental Evaluation

Our powerset CNN is built on the premise that the successful components of conventional CNNs are domain independent and only rely on the underlying concepts of shift and shift-equivariant convolutions. In particular, if we use only one-hop filters, our powerset CNN satisfies locality and compositionality. Thus, similar to image CNNs, it should be able to learn localized hierarchical features. To understand whether this is useful when applied to set function classification problems, we evaluate our powerset CNN architectures on three synthetic tasks and on two tasks based on real-world hypergraph data.

**Problem formulation** Intuitively, our set function classification task will require the models to learn to classify a collection of set functions sampled from some natural distributions. One such example would be to classify (hyper-)graphs coming from some underlying data distributions. Formally, the set function classification problem is characterized by a training set $\{(s^{(i)}, t^{(i)})\}_{i=1}^m \subseteq (\mathbb{R}^{2^N} \times \mathcal{C})$ composed of pairs (set function, label), as well as a test set. The learning task is to utilize the training set to learn a mapping from the space of set functions $\mathbb{R}^{2^N}$ to the label space $\mathcal{C} = \{1, \ldots, k\}$.

## 4.1 Synthetic Datasets

Unless stated otherwise, we consider the ground set $N = \{x_1, \ldots, x_n\}$ with $n = 10$, and sample $10,000$ set functions per class. We use $80\%$ of the samples for training, and the remaining $20\%$ for testing. We only use one random split per dataset. Given this, we generated the following three synthetic datasets, meant to illustrate specific applications of our framework.

**Spectral patterns** In order to obtain non-trivial classes of set functions, we define a sampling procedure based on the Fourier expansion associated with the shift $T_Q s = (s_{A \setminus Q})_{A \subseteq N}$. In particular, we sample Fourier sparse set functions, $s = F^{-1} \hat{s}$ with $\hat{s}$ sparse. We implement this by associating each target "class" with a collection of frequencies, and sample normally distributed Fourier coefficients for these frequencies. In our example, we defined four classes, where the Fourier support of the first and second class is obtained by randomly selecting roughly half of the frequencies. For the third class we use the entire spectrum, while for the fourth we use the frequencies that are either in both of class one's and class two's Fourier support, or in neither of them.

**$k$-junta classification** A $k$-junta [33] is a boolean function defined on $n$ variables $x_1, \ldots, x_n$ that only depends on $k$ of the variables: $x_{i_1}, \ldots, x_{i_k}$. In the same spirit, we call a set function a $k$-junta if its evaluations only depend on the presence or absence of $k$ of the $n$ elements of the ground set:

**Definition 5.** ($k$-junta) *A set function $s$ on the ground set $N$ is called a $k$-junta if there exists a subset $N' \subseteq N$, with $|N'| = k$, such that $s(A) = s(A \cap N')$, for all $A \subseteq N$.*

We generate a $k$-junta classification dataset by sampling random $k$-juntas for $k \in \{3, \ldots, 7\}$. We do so by utilizing the fact that shifting a set function by $\{x\}$ eliminates its dependency on $x$, i.e., for $A$ with $x \in A$ we have $(T_{\{x\}} s)_A = s_{A \setminus \{x\}} = (T_{\{x\}} s)_{A \setminus \{x\}}$ because $(A \setminus \{x\}) \setminus \{x\} = A \setminus \{x\}$. Therefore, sampling a random $k$-junta amounts to first sampling a random value for every subset $A \subseteq N$ and performing $n - k$ set shifts by randomly selected singleton sets.

**Submodularity classification** A set function $s$ is submodular if it satisfies the diminishing returns property

$$\forall A, B \subseteq N \text{ with } A \subseteq B \text{ and } \forall x \in N \setminus B : s_{A \cup \{x\}} - s_A \geq s_{B \cup \{x\}} - s_B. \tag{14}$$

In words, adding an element to a small subset increases the value of the set function at least as much as adding it to a larger subset. We construct a dataset comprised of submodular and "almost submodular" set functions. As examples of submodular functions we utilize coverage functions [26] (a subclass of submodular functions that allows for easy random generation). As examples of what we informally call "almost submodular" set functions here, we sample coverage functions and perturb them slightly to destroy the coverage property.

## 4.2 Real Datasets

Finally, we construct two classification tasks based on real hypergraph data. Reference [5] provides 19 real-world hypergraph datasets. Each dataset is a hypergraph evolving over time. An example is the DBLP coauthorship hypergraph in which vertices are authors and hyperedges are publications. In the following, we consider classification problems on subhypergraphs induced by vertex subsets of size ten. Each hypergraph is represented by its weight set function $s_A = 1$ if $A \in E$ and $s_A = 0$ otherwise.

**Definition 6.** (Induced Subhypergraph [6]) *Let $H = (V, E)$ be a hypergraph. The subset of vertices $V' \subseteq V$ induces a subhypergraph $H' = (V', E')$ with $E' = \{A \cap V' : \text{ for } A \in E \text{ and } A \cap V' \neq \emptyset\}$.*

**Domain classification** As we have multiple hypergraphs, an interesting question is whether it is possible to identify from which hypergraph a given subhypergraph of size ten was sampled, i.e., whether it is possible to distinguish the hypergraphs by considering only local interactions. Therefore, among the publicly available hypergraphs in [5] we only consider those containing at least 500 hyperedges of cardinality ten (namely, *DAWN*: 1159, *threads-stack-overflow*: 3070, *coauth-DBLP*: 6599, *coauth-MAG-History*: 1057, *coauth-MAG-Geology*: 7704, *congress-bills*: 2952). The *coauth*-hypergraphs are coauthorship hypergraphs, in *DAWN* the vertices are drugs and the hyperedges patients, in *threads-stack-overflow* the vertices are users and the hyperedges questions on threads on `stackoverflow.com` and in *congress-bills* the vertices are congresspersons and the hyperedges cosponsored bills. From those hypergraphs we sample all the subhypergraphs induced by the

hyperedges of size ten and assign the respective hypergraph of origin as class label. In addition to this dataset (*DOM6*), we create an easier version (*DOM4*) in which we only keep one of the coauthorship hypergraphs, namely *coauth-DBLP*.

**Simplicial closure** Reference [5] distinguishes between open and closed hyperedges (the latter are called simplices). A hyperedge is called open if its vertices in the 2-section (the graph obtained by making the vertices of every hyperedge a clique) of the hypergraph form a clique and it is not contained in any hyperedge in the hypergraph. On the other hand, a hyperedge is closed if it is contained in one or is one of the hyperedges of the hypergraph. We consider the following classification problem: For a given subhypergraph of ten vertices, determine whether its vertices form a closed hyperedge in the original hypergraph or not.

In order to obtain examples for closed hyperedges, we sample the subhypergraphs induced by the vertices of hyperedges of size ten and for open hyperedges we sample subhypergraphs induced by vertices of hyperedges of size nine extended by an additional vertex. In this way we construct two learning tasks. First, *CON10* in which we extend the nine-hyperedge by choosing the additional vertex such that the resulting hyperedge is open (2952 closed and 4000 open examples). Second, *COAUTH10* in which we randomly extend the size nine hyperedges (as many as there are closed ones) and use *coauth-DBLP* for training and *coauth-MAG-History* & *coauth-MAG-Geology* for testing.

## 4.3 Experimental Setup

**Baselines** As baselines we consider a multi-layer perceptron (MLP) [43] with two hidden layers of size 4096 and an appropriately chosen last layer and graph CNNs (GCNs) on the undirected $n$-dimensional hypercube. Every vertex of the hypercube corresponds to a subset and vertices are connected by an edge if their subsets only differ by one element. We evaluate graph convolutional layers based on the Laplacian shift [25] and based on the adjacency shift [44]. In both cases one layer does at most one hop.

**Our models** For our powerset CNNs (PCNs) we consider convolutional layers based on one-hop filters of two different convolutions: $(h * s)_A = h_\emptyset s_A + \sum_{x \in N} h_{\{x\}} s_{A \setminus \{x\}}$ and $(h \diamond s)_A = h_\emptyset s_A + \sum_{x \in N} h_{\{x\}} s_{A \cup \{x\}}$. For all types of convolutional layers we consider the following models: three convolutional layers followed by an MLP with one hidden layer of size 512 as illustrated before, a pooling layer after each convolutional layer followed by the MLP, and a pooling layer after each convolutional layer followed by an accumulation step (average of the features over all subsets) as in [18] followed by the MLP. For all models we use 32 output channels per convolutional layer and ReLU [32] non-linearities.

**Training** We train all models for 100 epochs (passes through the training data) using the Adam optimizer [24] with initial learning rate 0.001 and an exponential learning rate decay factor of 0.95. The learning rate decays after every epoch. We use batches of size 128 and the cross entropy loss. All our experiments were run on a server with an Intel(R) Xeon(R) CPU @ 2.00GHz with four NVIDIA Tesla T4 GPUs. Mean and standard deviation are obtained by running each experiment 20 times.

## 4.4 Results

Our results are summarized in Table 2. We report the test classification accuracy in percentages (for models that converged).

**Discussion** Table 2 shows that in the *synthetic tasks* the powerset convolutional models ($*$-PCNs) tend to outperform the baselines with the exception of $A$-GCNs, which are based on the adjacency graph shift on the undirected hypercube. In fact, the set of $A$-convolutional filters parametrized by our $A$-GCNs is the subset of the powerset convolutional filters associated with the symmetric difference shift (6) obtained by constraining all filter coefficients for one-element sets to be equal: $h_{\{x_i\}} = c$ with $c \in \mathbb{R}$, for all $i \in \{1, \ldots, n\}$. Therefore, it is no surprise that the $A$-GCNs perform well. In contrast, the restrictions placed on the filters of $L$-GCN are stronger, since [25] replaces the one-hop Laplacian convolution $(\theta_0 I + \theta_1 (L - I))x$ (in Chebyshev basis) with $\theta(2I - L)x$ by setting $\theta = \theta_0 = -\theta_1$.

An analogous trend is not as clearly visible in the tasks derived from *real hypergraph data*. In these tasks, the graph CNNs seem to be either more robust to noisy data, or, to benefit from their permutation equivariance properties. The robustness as well as the permutation equivariance can

| | Patterns | k-Junta | Submod. | COAUTH10 | CON10 | DOM4 | DOM6 |
|---|---|---|---|---|---|---|---|
| **Baselines** | | | | | | | |
| MLP | $46.8 \pm 3.9$ | $43.2 \pm 2.5$ | - | $80.7 \pm 0.2$ | $66.1 \pm 1.8$ | $93.6 \pm 0.2$ | $71.1 \pm 0.3$ |
| L-GCN | $52.5 \pm 0.9$ | $69.3 \pm 2.8$ | - | $\mathbf{84.7 \pm 0.9}$ | $\mathbf{67.2 \pm 1.8}$ | $\mathbf{96.0 \pm 0.2}$ | $\mathbf{73.7 \pm 0.4}$ |
| L-GCN pool | $45.0 \pm 1.0$ | $60.9 \pm 1.5$ | - | $83.2 \pm 0.7$ | $65.7 \pm 1.0$ | $93.2 \pm 1.1$ | $71.7 \pm 0.5$ |
| L-GCN pool avg. | $42.1 \pm 0.3$ | $64.3 \pm 2.2$ | $82.2 \pm 0.4$ | $56.8 \pm 1.1$ | $64.1 \pm 1.7$ | $88.4 \pm 0.3$ | $62.8 \pm 0.4$ |
| A-GCN | $\mathbf{65.5 \pm 0.9}$ | $95.8 \pm 2.7$ | - | $80.5 \pm 0.7$ | $64.9 \pm 1.8$ | $93.9 \pm 0.3$ | $69.1 \pm 0.5$ |
| A-GCN pool | $56.9 \pm 2.2$ | $91.9 \pm 2.1$ | $\mathbf{89.8 \pm 1.8}$ | $84.1 \pm 0.6$ | $66.0 \pm 1.6$ | $93.8 \pm 0.3$ | $70.7 \pm 0.4$ |
| A-GCN pool avg. | $54.8 \pm 0.9$ | $\mathbf{95.8 \pm 1.1}$ | $84.8 \pm 1.9$ | $64.8 \pm 1.1$ | $65.4 \pm 0.7$ | $92.7 \pm 0.6$ | $67.9 \pm 0.3$ |
| **Proposed models** | | | | | | | |
| ∗-PCN | $\mathbf{88.5 \pm 4.3}$ | $97.2 \pm 2.3$ | $\mathbf{88.6 \pm 0.4}$ | $80.6 \pm 0.7$ | $62.8 \pm 2.9$ | $94.1 \pm 0.3$ | $70.5 \pm 0.3$ |
| ∗-PCN pool | $80.9 \pm 0.9$ | $96.0 \pm 1.6$ | $85.1 \pm 1.8$ | $82.6 \pm 0.4$ | $62.9 \pm 2.0$ | $94.0 \pm 0.3$ | $70.2 \pm 0.5$ |
| ∗-PCN pool avg. | $75.9 \pm 1.9$ | $96.5 \pm 0.6$ | $87.0 \pm 1.6$ | $80.6 \pm 0.5$ | $63.4 \pm 3.5$ | $94.4 \pm 0.3$ | $73.0 \pm 0.3$ |
| ◇-PCN | - | $97.5 \pm 1.4$ | - | $83.6 \pm 0.4$ | $\mathbf{68.7 \pm 1.3}$ | $93.7 \pm 0.2$ | $69.9 \pm 0.3$ |
| ◇-PCN pool | - | $96.4 \pm 1.7$ | - | $\mathbf{84.8 \pm 0.3}$ | $68.2 \pm 0.8$ | $93.6 \pm 0.3$ | $70.3 \pm 0.4$ |
| ◇-PCN pool avg. | $54.8 \pm 1.9$ | $96.6 \pm 0.7$ | $80.9 \pm 2.9$ | $83.3 \pm 0.5$ | $67.0 \pm 2.0$ | $\mathbf{94.8 \pm 0.3}$ | $\mathbf{73.5 \pm 0.5}$ |

Table 2: Results of the experimental evaluation in terms of test classification accuracy (percentage). The first three columns contain the results from the synthetic experiments and the last four columns the results from the hypergraph experiments. The best-performing model from the corresponding category is in bold.

be attributed to the graph one-hop filters being omnidirectional. On the other hand, the powerset one-hop filters are $n$-directional. Thus, they are sensitive to hypergraph isomorphy, i.e., hypergraphs with same connectivity structure but different vertex ordering are being processed differently.

**Pooling** Interestingly, while reducing the hidden state by a factor of two after every convolutional layer, pooling in most cases only slightly decreases the accuracy of the PCNs in the synthetic tasks and has no impact in the other tasks. Also the influence of pooling on the $A$-GCN is more similar to the behavior of PCNs than the one for the $L$-GCN.

**Equivariance** Finally, we compare models having a shift-invariant convolutional part (suffix "pool avg.") with models having a shift-equivariant convolutional part (suffix "pool") models. The difference between these models is that the invariant ones have an accumulation step before the MLP resulting in (a) the inputs to the MLP being invariant w.r.t. the shift corresponding to the specific convolutions used and (b) the MLP having much fewer parameters in its hidden layer ($32 \cdot 512$ instead of $2^{10} \cdot 32 \cdot 512$). For the PCNs the effect of the accumulation step appears to be task dependent. For instance, in $k$-*Junta*, *Submod.*, *DOM4* and *DOM6* it is largely beneficial, and in the others it slightly disadvantageous. Similarly, for the GCNs the accumulation step is beneficial in $k$-*Junta* and disadvantageous in *COAUTH10*. A possible cause is that the resulting models are not expressive enough due to the lack of parameters.

**Complexity analysis** Consider a powerset convolutional layer (11) with $n_c$ input channels and $n_f$ output channels. Using $k$-hop filters, the layer is parametrized by $n_p = n_f + n_c n_f \sum_{i=0}^{k} \binom{n}{i}$ parameters ($n_f$ bias terms plus $n_c n_f \sum_{i=0}^{k} \binom{n}{i}$ filtering coefficients). Convolution is done efficiently in the Fourier domain, i.e., $h * s = F^{-1}(\mathrm{diag}(\bar{F}h)Fs)$, which requires $\frac{3}{2} n 2^n + 2^n$ operations and $2^n$ floats of memory [36]. Thus, forward as well as backward pass require $\Theta(n_c n_f n 2^n)$ operations and $\Theta(n_c 2^n + n_f 2^n + n_p)$ floats of memory[2]. The hypercube graph convolutional layers are a special case of powerset convolutional layers. Hence, they are in the same complexity class. A $k$-hop graph convolutional layer requires $n_f + n_c n_f (k + 1)$ parameters.

## 5   Related Work

Our work is at the intersection of geometric deep learning, generalized signal processing and set function learning. Since each of these areas is broad, due to space limitations, we will only review the work that is most closely related to ours.

**Deep learning** Geometric deep learners [10] can be broadly categorized into convolution-based approaches [9, 12, 13, 16, 25, 55] and message-passing-based approaches [18, 47, 50]. The latter assign a hidden state to each element of the index domain (e.g., to each vertex in a graph) and make use of a message passing protocol to learn representations in a finite amount of communication steps.

Reference [18] points out that graph CNNs are a subclass of message passing / graph neural networks (MPNNs). References [9, 16, 25] utilize the spectral analysis of the graph Laplacian [51] to define graph convolutions, while [55] makes use of the adjacency shift based convolution [44]. Similarly, [12, 13] utilize group convolutions [53] with desirable equivariances.

In a similar vein, in this work we utilize the recently proposed powerset convolutions [36] as the foundation of a generalized CNN. With respect to the latter reference, which provides the theoretical foundation for powerset convolutions, our contributions are an analysis of the resulting filters from a pattern matching perspective, to define its exact instantiations and applications in the context of neural networks, as well as to show that these operations are practically relevant for various tasks.

**Signal processing** Set function signal processing [36] is an instantiation of algebraic signal processing (ASP) [38] on the powerset domain. ASP provides a theoretical framework for deriving a complete set of basic signal processing concepts, including convolution, for novel index domains, using as starting point a chosen shift to which convolutions should be equivariant. To date the approach was used for index domains including graphs [34, 44, 45], powersets (set functions) [36], meet/join lattices [37, 61], and a collection of more regular domains, e.g., [39, 46, 49].

Additionally, there are spectral approaches such as [51] for graphs and [15, 33] for set functions (or, equivalently, pseudo-boolean functions), that utilize analogues of the Fourier transform to port spectral analysis and other signal processing methods to novel domains.

**Set function learning** In contrast to the set function classification problems considered in this work, most of existing set function learning is concerned with completing a single partially observed set function [2–4, 7, 11, 30, 54, 56, 63]. In this context, traditional methods [2–4, 11, 30, 56] mainly differ in the way how the class of considered set functions is restricted in order to be manageable. E.g., [54] does this by considering Walsh-Hadamard-sparse (= Fourier sparse) set functions. Recent approaches [7, 17, 31, 57, 60, 63] leverage deep learning. Reference [7] proposes a neural architecture for learning submodular functions and [31, 63] propose architectures for learning multi-set functions (i.e., permutation-invariant sequence functions). References [17, 57] introduce differentiable layers that allow for backpropagation through the minimizer or maximizer of a submodular optimization problem respectively and, thus, for learning submodular set functions. Similarly, [60] proposes a differentiable layer for learning boolean functions.

# 6 Conclusion

We introduced a convolutional neural network architecture for powerset data. We did so by utilizing novel powerset convolutions and introducing powerset pooling layers. The powerset convolutions used stem from algebraic signal processing theory [38], a theoretical framework for porting signal processing to novel domains. Therefore, we hope that our method-driven approach can be used to specialize deep learning to other domains as well. We conclude with challenges and future directions.

**Lack of data** We argue that certain success components of deep learning are domain independent and our experimental results empirically support this claim to a certain degree. However, one cannot neglect the fact that data abundance is one of these success components and, for the supervised learning problems on set functions considered in this paper, one that is currently lacking.

**Computational complexity** As evident from our complexity analysis and [29], the proposed methodology is feasible only up to about $n = 30$ using modern multicore systems. This is caused by the fact that set functions are exponentially large objects. If one would like to scale our approach to larger ground sets, e.g., to support semisupervised learning on graphs or hypergraphs where there is enough data available, one should either devise methods to preserve the sparsity of the respective set function representations while filtering, pooling and applying non-linear functions, or, leverage techniques for NN dimension reduction like [21].

# Acknowledgements

We thank Max Horn for insightful discussions and his extensive feedback, and Razvan Pascanu for feedback on an earlier draft. This project has received funding from the European Research Council (ERC) under the European Union's Horizon 2020 research and innovation programme (grant agreement No 805223).

## Footnotes

[1]Sample implementations are provided at `https://github.com/chrislybaer/Powerset-CNN`.

[2]The derivation of these results is provided in the supplementary material.

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
