[Supplementary Material]

# Powerset Convolutional Neural Networks
## Supplementary Material

**Chris Wendler**
Department of Computer Science
ETH Zurich, Switzerland
chris.wendler@inf.ethz.ch

**Dan Alistarh**
IST Austria
dan.alistarh@ist.ac.at

**Markus Püschel**
Department of Computer Science
ETH Zurich, Switzerland
pueschel@inf.ethz.ch

## 1 Complexity Analysis

**Definition 1.** (Powerset convolutional layer) *A powerset convolutional layer is defined as follows:*

1. *The input is given by $n_c$ set functions $\mathbf{s} = (s^{(1)}, \ldots, s^{(n_c)}) \in \mathbb{R}^{2^N \times n_c}$ ;*

2. *The output is given by $n_f$ set functions $\mathbf{t} = L_\Gamma(\mathbf{s}) = (t^{(1)}, \ldots, t^{(n_f)}) \in \mathbb{R}^{2^N \times n_f}$;*

3. *The layer applies a bank of set function filters $\Gamma = (h^{(i,j)})_{i,j}$, with $i \in \{1, \ldots, n_c\}$ and $j \in \{1, \ldots, n_f\}$, and a point-wise non-linearity $\sigma$ resulting in*

$$t_A^{(j)} = \sigma(\sum_{i=1}^{n_c} (h^{(i,j)} * s^{(i)})_A). \tag{1}$$

While the provided Tensorflow is prototypical, our analysis assumes fast implementations. Consider a powerset convolutional layer (1) with $n_c$ input channels and $n_f$ output channels. Convolution is done efficiently in the Fourier domain, i.e., $h * s = F^{-1}(\text{diag}(\bar{F}h)Fs)$, which requires $\frac{3}{2}n2^n + 2^n$ operations and $2^n$ floats of memory due to the Kronecker-structure of the frequency response $\bar{F}$ and Fourier transform $F$.

**Operations** A *forward pass* requires $n_c n_f(\frac{3}{2}n2^n + 2^n)$ operations, as for each input- and output channel one convolution is performed. For the *backward pass* the Jacobian $\frac{\partial t^{(j)}}{\partial h^{(i,j)}} = \text{diag}((\frac{\partial \sigma}{\partial x_A^{(j)}})_{A \subseteq N})I_{2^n}F^{-1}\text{diag}(\hat{s}^{(i)})\bar{F}$ is multiplied by the $1 \times 2^n$ accumulated Jacobian of the consecutive layers $\triangle_{t^{(j)}}$ from the left requiring $n2^n + 2^{n+1}$ operations. Doing this for all filters $h^{(i,j)}$ yields $n_c n_f(n2^n + 2^{n+1})$. Similarly, computing all $\triangle_{t^{(j)}} \frac{\partial t^{(j)}}{\partial s^{(i)}}$ requires $n_c n_f(n2^n + 2^{n+1})$ operations. Therefore, $\frac{\partial t}{\partial \mathbf{s}}$ and $\frac{\partial t}{\partial \Gamma}$ require $2n_c n_f(n2^n + 2^{n+1})$ operations.

**Memory** A *forward pass* requires $n_c 2^n + n_f 2^n + \#(params.)$ floats and a *backward pass* $n_f 2^n + n_c 2^n + \#(params.)$ floats.

**Parameters** Using $k$-hop filters, a layer requires $n_f$ bias terms and $n_c n_f \sum_{i=0}^k \binom{n}{i}$ coefficients.

**Baselines**  Graph convolutional layers for the undirected hypercube graph are a special case of powerset convolutional layers. Hence, they are in the same complexity class. A $k$-hop graph convolutional layer requires $n_f + n_c n_f (k + 1)$ parameters.