[Reviews · NeurIPS 2019]

Reviewer 1



The paper is clear, well motivated, well written and overall very pleasant to read. The proposed method is new and interesting and will probably remain as a baseline for future work in the topic of deep learning on set functions. Some remarks: - l95 are defined cut and association score but they are not used in the following of the paper. - What is the norm on s:2^N -> R (used e.g. l 99-100)? - l194: the sentence "hypergraphs are represented by their weight set function (with unit weights)" is unclear - l232: two different models are proposed and benchmarked, a discussion about their difference would be very welcomed - l232: it would nice to have the number of parameters for the different proposed models - l302: "(i.e. Fourier sparse)" instead of "(= Fourier sparse)". - In the case of set functions, the complexity of the related algorithm may become an issue, how does the proposed algorithm compared time wise with the other benchmark methods? An in-depth discussion about applicability of the algorithm would be welcomed. * Bibliography: many references are incomplete, e.g. [4] is COLT 2012, [12] is ICLR 2018, etc.

Reviewer 2



The paper considers an interesting and relatively novel machine learning problem, learning on set functions. It proposes a new architecture called powerset convolutional NN tailed for this problem. The proposed model is sted from solid theoretical analysis and gets reasonable good experiment results. In every dimension of "originality, quality, clarity, and significance", I think this paper reaches the bar of NeurIPS. Section 2 is my favorite part of the paper. It is very clear and constructs the powerset convolutions by analyzing the shift-invariant and from easy one to more complex ones. I also like Table 1 a lot. It is really good to see the corresponds between convolutions and their shifts. The author did an interesting analysis of convolutional pattern matching. I am wondering since powerset convolution and graph convolution has different pattern matching behavior comparing to 1D or 2D convolution, does it finally cause some behavior difference between CNN and GCN/Powerset CNN. The paper also has limitations. First, its set function representation is very costive. It uses R^{2^N} vector represents a set function on a set with N elements. Thus it is hard to apply this method to a large set. It may restrict its use in real applications. Second, the experiments conducted on relatively small datasets. For example, even in the real dataset, results are about classification on subhypergraph of size 10. While graph neural network can handle graph with a much larger size.

Reviewer 3



The authors built their work on top of "A discrete signal processing framework for set functions" where powerset convolutions were defined, adding powerset pooling operations and defining powerset convolutional neural networks that can be used to classify set functions. The authors provided a detailed analysis of the kind of patterns that powerset convolutions are sensitive to from a pattern matching perspective, and defined their implementation. The authors recognize the exponential growth of complexity O(n2^n) and that to scale their approach to larger ground sets, which limits the applicability of the current method. The empirical results show that the powerset CNNs perform similarly to the baselines on both the synthetic and real datasets, maybe the tasks chosen are too small or well suited to showcase the proposed powerset CNNs. The authors recognizes the lack of datasets containing set functions well suited for their method, however the current set of experiments weakens the argument than powerset CNNs can handle set functions better than graph-convolutional baselines. If the proposed powerset CNNs could be applied to theorem proving or to transformations of boolean formulas, then it could show their relevance and applicability. The paper provides clear definition of all the concepts used to define powerset CNNs, including the different shifts and invariances of the different convolutions and poolings operations. Minor comments: - 19 according parameter -> appropriate parameter - 307 utilizing novel powerset convolutions -> utilizing recent powerset convolutions ------------- POST FEEDBACK --------------- After reviewing the author feedback, I've updated my score to 5, I think the work could be accepted but is not clear how applicable it would be given current restrictions ground-set size = 30, and lack of clear examples where powerset CNNs are better than graph CNNS. The way the contributions of the paper are described need to be updated to clearly reflects the contributions of the paper and what is previous work.

[Author Response · NeurIPS 2019]

We thank the reviewers for their comments. We will incorporate all points and suggested clarifications. We assume a
ground set of size $n$. Note that set functions are inherently $2^n$-dimensional. The goal of the paper is to provide a novel,
mathematically sound, CNN architecture with prototypical evaluation that the community can build on. We first answer
a common question about the complexity; we will include the detailed derivation in the final version.
**Complexity analysis** We consider a powerset convolutional layer with $n_c$ input channels and $n_f$ output channels.
Convolution is done efficiently in the Fourier domain, i.e., $h * s = F^{-1}(\text{diag}(\bar{F}h)Fs)$, which requires $\frac{3}{2}n2^n + 2^n$
operations and $2^n$ floats of memory.
*a. Operations:* forward pass: $n_c n_f(\frac{3}{2}n2^n + 2^n)$ operations, backward pass: $2n_c n_f(n2^n + 2^{n+1})$ operations.
*b. Memory:* forward pass: $n_c 2^n + n_f 2^n + \#(params.)$ floats, backward pass: $n_f 2^n + n_c 2^n + \#(params.)$ floats.
*c. Parameters:* Using $k$-hop filters, a layer requires $n_f$ bias terms and $n_c n_f \sum_{i=0}^{k} \binom{n}{i}$ filtering coefficients.
*d. Baseline:* Graph convolutional layers for a hypercube are a special case of powerset convolutional layers (l.250-252).
Hence, they are in the same complexity class. A $k$-hop graph convolutional layer requires $n_f + n_c n_f(k+1)$ parameters.
**Improvements** Using modern GPUs, ground sets up to size $n \equiv 30$ are feasible [A]. Our TensorFlow implementation
is a prototype meant to demonstrate viability, and thus has limited efficiency. Future work could leverage techniques for
NN dimension reduction, e.g. [B], to scale powerset CNNs to larger domains.
**Reviewer 1**
Q. *What is the norm on $s : 2^N \to \mathbb{R}$ (used e.g. l.99-100)?*
R. $\|s\| = (\sum_{A \subseteq N} s_A^2)^{1/2}$ .
Q. *What is the difference between the two proposed models (l.232)?*
R. $*$-PCNs are shift-equivariant w.r.t. $s \mapsto (s_{A \setminus Q})_{A \subseteq N}$ and $\diamond$-PCNs w.r.t. its dual shift, i.e., $s \mapsto (s_{A \cup Q})_{A \subseteq N}$.
Q. *It would be nice to see a benchmark with more than one-hop filters if doable.*
R. As we are filtering in Fourier domain even $n$-hops are doable. We ran the benchmark using 2-hop filters and saw
only a small improvement only in some cases. This is likely due to the small scale of our prototypical experiments.
Q. *How would the proposed method specialize to graphs and how would it compare to classical GNNs?*
R. A weighted graph is a special set function with values only on the two element sets (the edges). Using, e.g.,
$(h * s)_A = \sum_{Q \subseteq N} h_Q s_{A \setminus Q}$, the powerset CNN would create nonzero values for larger sets (of nodes), i.e., turning it
into an edge-weighted hypergraph, increasing the dimension of the data, in contrast to graph NNs.
**Reviewer 2**
Q. *Real applications, e.g., in the scope of sensor- or ad-placement, would significantly strengthen this work.*
R. These two tasks are subset-selection tasks, in which a set function serves as a tool to assess the quality of subsets,
e.g., by assigning a score to each subset. As a consequence, the set function problems considered in these area are 1.
finding the subset with the highest score subject to some constraints [17, 24] and 2. learning the scoring function [45].
Problem 1 is not a learning problem and Problem 2 is a transductive learning task. Therefore, the proposed method
does not directly apply, and instead would require to be specialized to the transductive setting (if possible).
**Reviewer 3**
Q. *What are the novel contributions of this paper, and what is prior work [31]?*
R. Convolution and associated Fourier transforms were defined in [31] as cited. However, we are the first to extend
these results to design and apply powerset CNNs. This includes the definition of convolutional and pooling layers, the
analysis of patterns matched, and a prototypical implementation and evaluation to show viability. The contribution is
somewhat similar to graph CNNs (e.g. [9]), which built on long existing results from algebraic graph theory.
Q. *Showing that powerset CNNs can solve tasks defined on set functions better than the baselines, and that they are*
*indeed superior to graph-convolutions for those tasks.*
R. To our best knowledge there is no prior work on set function classification. Our baseline—viewing them as data
indexed by an undirected hypercube graph—is thus also novel. These graph convolutions are a small subset of the
powerset convolutions based on the symmetric shift in Equation (6), for which we did not include experiments. We
showed, prototypically, that the directed shifts (adding or subtracting an element) can yield improvements and are thus
viable for applications.
Q. *Could the proposed powerset CNNs be applied to convolution-deconvolution networks that would allow set function*
*learning and transformation?*
R. (Not in the paper) We successfully trained fully convolutional 1-hop and $n$-hop powerset CNNs to solve a similar
task, namely, to transform probability mass functions $p$, with $p_A = p_x$ for $A = \{x\}$ and $p_A = 0$ otherwise, to their
associated probability measures $P_A = \sum_{x \in A} p_x$. We claim that it is possible to utilize our layers within a variational
autoencoder and, thus, to learn to sample set functions from a target distribution. The challenges in doing so are 1. to
define a bottleneck, e.g., through pooling, 2. a corresponding scheme to undo the dimensionality reduction, and 3. to
find training data. These autoencoders could find application in simulation frameworks used in combinatorial auctions
[13] and to generate submodular functions. We are not sure whether such an architecture would be suitable for Boolean
function synthesis or transformation, as it would require truth-tables as inputs/outputs rather than Boolean expressions.

**References** [A] Yi Lu: "Practical tera-scale Walsh-Hadamard Transform." In FTC 2016.    [B] Hackel et al.: "Inference,
Learning and Attention Mechanisms that Exploit and Preserve Sparsity in CNNs." In GCPR 2018.


[Meta-Review · NeurIPS 2019]

The authors present a neural network architecture for set functions, i.e. to identify a subset within a larger set. The authors provide a clear introduction to the problem in terms of convolutional operators and design CNN architectures on top of [1] through the addition of pooling operations for addressing the set function problem. This work tests this method on 3 synthetic problems as well several real world problems. The resulting networks performed competitively with baseline graph convolutional networks although they were outperformed slightly on subsets of tasks. The reviewers greatly appreciated the presentation of the work as the ideas were well motivated, the explanations were clear and the overall presentation were organized. Reviewers commented on the fact that the experiments were conducted on relatively small datasets. This reflected both the lack of good training data but also the scaling limitations of the proposed method. Indeed, as R2 suggests, it would be great for the authors to demonstrate the applicability and utility of this method on real world data as suggested in Section 2.1. The lack of experiments demonstrating a strong utility for the method on real world data is concerning and a detraction of the paper. The authors are strongly encouraged to pursue research in this direction. That said, given the clarity of presentation and the strong motivation that this is an important problem, the meta-reviewer is favorable to accepting this paper into this conference provided the authors address all of the points and remarks provided by the reviewers. This paper provides an important touchpoint on an interesting machine learning problem. [1] Markus PĆ¼schel. A discrete signal processing framework for set functions. In Proc. International Conference on Acoustics, Speech, and Signal Processing (ICASSP), 2018.